# Clinical Outcomes Following Proton and Photon Stereotactic Body Radiation Therapy for Early-Stage Lung Cancer

**DOI:** 10.3390/cancers14174152

**Published:** 2022-08-27

**Authors:** Bong Kyung Bae, Kyungmi Yang, Jae Myung Noh, Hongryull Pyo, Yong Chan Ahn

**Affiliations:** Department of Radiation Oncology, Samsung Medical Center, Sungkyunkwan University School of Medicine, Seoul 06351, Korea

**Keywords:** early-stage lung cancer, stereotactic body radiation therapy, proton beam, photon beam

## Abstract

**Simple Summary:**

The current study reports the clinical outcomes of proton and photon stereotactic body radiation therapy (SBRT) for early-stage lung cancer. Out of 202 patients who met the inclusion criteria, 34 received proton SBRT and 168 received photon SBRT. Patients at high risk of developing post-SBRT radiation pneumonitis tended to receive proton SBRT. Oncologic outcomes and toxicity profiles were comparable between treatment modalities. Proton SBRT could be considered for patients with high risk of radiation pneumonitis.

**Abstract:**

We aimed to report the clinical outcomes following stereotactic body radiation therapy (SBRT) using photon or proton equipment in early-stage lung cancer. We retrospectively reviewed 202 cT1-2N0M0 lung cancer patients who underwent SBRT with 60 Gy in four consecutive fractions between 2010 and 2019 at our institution: 168 photon SBRT and 34 proton SBRT. Patients who underwent proton SBRT had relatively poor baseline lung condition compared to those who underwent photon SBRT. Clinical outcomes were comparable between treatment modalities: 5-year local control (90.8% vs. 83.6%, *p* = 0.602); progression-free survival (61.6% vs. 57.8%, *p* = 0.370); overall survival (51.7% vs. 51.9%, *p* = 0.475); and cause-specific survival (70.3% vs. 62.6%, *p* = 0.618). There was no statistically significant difference in grade ≥ 2 toxicities: radiation pneumonitis (19.6% vs. 26.4%, *p* = 0.371); musculoskeletal (13.7% vs. 5.9%, *p* = 0.264); and skin (3.6% vs. 0.0%, *p* = 0.604). In the binary logistic regression analysis of grade ≥3 radiation pneumonitis, poor performance status and poor baseline diffusion capacity of lung for carbon monoxide were significant. To summarize, though patients with high risk of developing lung toxicity underwent proton SBRT more frequently, the SBRT techniques resulted in comparable oncologic outcomes with similar toxicity profiles. Proton SBRT could be considered for patients at high risk of radiation pneumonitis.

## 1. Introduction

The incidence and mortality of lung cancer are continuously rising [1]. Based on the global cancer statistics 2020, lung cancer is the most common cancer and the leading cause of cancer deaths in males, and the third most common cancer and second most common cause of cancer deaths in females, respectively [2]. The gold standard management of early-stage (cT1-2N0M0) non-small cell lung cancer (NSCLC) is complete surgical resection with lobectomy [3]. However, more than two-thirds of the patients are 65 years or older, usually having multiple comorbidities, which frequently makes these patients medically unfit for surgery [4]. Therefore, a non-surgical approach, such as stereotactic body radiation therapy (SBRT), has been generally accepted as the standard of care as the surgical alternative [5,6].

The Bragg-peak phenomenon, the unique physical property of proton beam therapy, can provide a dosimetric advantage over photon beam therapy in several clinical situations [7]. This physical advantage of a proton beam, however, is known to exert an insignificant clinical benefit in the lung SBRT setting, and as a consequence, proton SBRT has not been widely recommended over photon SBRT [8,9,10,11]. There are a few clinical situations in which the physical benefit of proton SBRT may be practically effective by diminishing the risk of normal organ damage: tumors located immediately adjacent to critical central normal organs, or peripherally located and close to the chest wall; large tumors; patients with pre-existing lung disease, especially interstitial lung disease (ILD) and chronic obstructive pulmonary disease (COPD); and poor baseline pulmonary function test (PFT) results [10,12,13,14].

However, there is a paucity of clinical data which compare proton SBRT and photon SBRT in treating early-stage lung cancer patients. There was only one phase II study that compared proton SBRT and photon SBRT for medically inoperable, high risk NSCLC patients, which, however, was closed early because of poor patient accrual [15]. In this study, we intended to report the clinical outcomes following either photon or proton SBRT in early-stage lung cancer patients, which we believe was the largest clinical study in this context from a single tertiary institution under a consistent treatment policy.

## 2. Materials and Methods

### 2.1. Study Design

With the approval by Institutional Review Board of Samsung Medical Center (SMC IRB 2022-04-017-001), we retrospectively reviewed the medical records of the patients who underwent SBRT, either by photon or proton, for cT1-2N0M0 NSCLC by the 8^th^ edition AJCC staging system [16] between January 2010 and December 2019 at the authors’ institution. Operability of a certain patient was to be discussed at our regular multidisciplinary conference whenever there was any ambiguity. Patients were determined to be “medically inoperable” if they had one or more of the following: predicted FEV1 < 40%; predicted postoperative FEV1 < 30%; predicted DLCO < 40%; severe pulmonary hypertension; diabetes mellitus with end organ damage; severe cerebral, cardiovascular, or peripheral vascular disease; or severe chronic heart disease. Medically inoperable patients or patients who refused surgery were potential candidates for SBRT. In principle, we intended to include those with cyto-pathologic confirmation of lung cancer. However, there was significant proportion of patients in whom cyto-pathologic confirmation was very difficult or even risky due to poor underlying lung conditions. These patients were also included in the current study on the condition that the findings of two or more consecutive chest computerized tomography (CT) scans at 3–4 month intervals plus at least one 5-flourodeoxyglucose positron emission tomography and CT (PET-CT) were highly suggestive of primary lung cancer. The patients with previous history of malignancy other than lung cancer, double primary cancer at the time of diagnosis, cT3 or higher stage, previous history of radiation therapy, or dose schedule other than 60 Gy in four fractions were excluded from the current analysis.

### 2.2. Stereotactic Body Radiation Therapy

The proton therapy equipment was installed in March of 2016 at the authors’ institution, and soon started to be used in limited clinical situations. The first lung proton SBRT procedure started in January of 2017, before which all lung SBRT procedures were undertaken by photon equipment. After January of 2017, the candidates for lung SBRT were allocated either to photon or proton equipment mainly considering the pulmonary functional status of each patient in addition to the availability of the equipment and consequent expected treatment delay. As a consequence, proton SBRT tended to be more frequently allocated to the patients who were assumed to be at higher risk of post-SBRT radiation pneumonitis (underlying lung disease such as COPD or ILD, or poor baseline PFT results).

The general concepts of the patient preparation, target delineation, and treatment planning in lung SBRT were previously reported [17], and the same policies and dose schedules were applied to all patients. All patients were immobilized in the supine position, and the planning CT scans with 2.5 mm slice thickness were acquired while the patients were breathing according to the pre-recorded respiratory curve on each patient, which was displayed on a goggle. The 4D-CT images were reconstructed and binned into 10 respiratory phases with 0% representing end-inhalation and 50% representing end-exhalation. For the patients who showed large respiratory motion amplitude in planning CT and assigned to proton SBRT, re-simulation with respiratory motion control, such as deep inspiration breath holding, was considered and applied at the treating physicians’ discretion. The gross tumor volume (GTV) of each respiratory phase was contoured and summed to create the internal target volume (ITV). The clinical target volume (CTV) was generated by expanding 5 mm in all directions from the ITV, and the planning target volume (PTV) was generated with a 0–5 mm margin around the CTV considering the surrounding normal structures and anatomic barriers. SBRT was planned based on the average intensity projection image set. The dose was prescribed at the calculation point within the ITV, so that 95% of the PTV received ≥90% of the prescribed dose. 

Four RT techniques were applied for SBRT in the current study. To those who underwent photon SBRT, two RT techniques were used based on the availability of equipment throughout the study period: multi-port three-dimensional conformal radiation therapy (3D-CRT) was the main technique during the early part of the study, which was mostly replaced later with intensity modulated radiation therapy (IMRT) by volumetric arc modulation. To those who underwent photon SBRT, either passive scattering proton therapy or intensity modulated proton therapy with scanned beams was utilized.

### 2.3. Assessments

All patients were recommended to visit our follow-up clinic 1 month after SBRT completion with a chest CT, and every 3–4-months thereafter with PET-CT and chest CT alternatingly. Local control (LC) was defined as absence of local tumor progression (evident increase in primary tumor size). The durations of event-free intervals were defined as the intervals between the start of SBRT and the occurrence of corresponding events (treatment failures included local, regional, or distant failure, or death). The treatment-related toxicities were evaluated according to the Common Terminology Criteria for Adverse Events, version 3.0. Grade 2 or higher toxic events of the lung, esophagus, heart, musculoskeletal system, and skin were recorded.

### 2.4. Statistical Analysis

The Chi-square test or Fisher’s exact test was performed to compare the categorical variables, and the *t*-test was to compare the continuous variables between groups. The survivals were calculated using the Kaplan–Meier method, and the comparisons between groups were performed using the log-rank test. The binary logistic regression analysis was performed to determine the factors associated with grade 3 or higher radiation pneumonitis. The factors with *p*-value < 0.10 in the univariable analysis entered the multivariable analysis. The final multivariable model was determined using a backward variable selection method with elimination criteria of 0.05. For subgroup analysis, propensity score matching was performed in order to minimize the interference by the covariates of T stage, COPD, ILD, baseline FEV1, and baseline DLCO. Matching was carried out with the ratio of 1:2, the nearest-neighbor method, and the caliper distance of 0.2. A *p*-value of <0.05 was considered statistically significant. The statistical analyses were performed using the IBM SPSS Statistical Software version 27.0 (IBM, Inc., Armonk, NY, USA) or R software version 4.1.2 (R Foundation for Statistical Computing, Vienna, Austria; http://www.r-project.org, accessed on 4 April 2022).

## 3. Results

### 3.1. Patient and Treatment Characteristics

Out of 404 SBRTs during the study period, 202 patients met the inclusion criteria of the current study (Table 1). The median age of all patients at the time of SBRT was 75.0 years, and the vast majority were male (161 patients, 79.7%); had good general performance status by Eastern Cooperative Oncology Group performance status (ECOG PS) 0–1 (167 patients, 82.7%); were of medically inoperable status (155 patients, 76.7%); and had cT1 tumors (170 patients, 84.2%). Adenocarcinoma was most frequent (77 patients, 38.1%) histologic type, followed by squamous cell carcinoma (47 patients, 23.3%). All SBRT procedures prior to January of 2017 were delivered with a photon beam (80 patients), whereas those after January of 2017 were delivered with either a photon beam (88 patients) or a proton beam (34 patients). The median follow-up duration of all patients was 2.6 years: 2.7 years for photon SBRT and 2.2 years for proton SBRT. 

The patients who underwent proton SBRT generally had relatively poor baseline lung condition when compared to those who underwent photon SBRT: COPD 70.6% vs. 36.3% (*p* < 0.001); the mean baseline predicted forced expiratory volumes in 1 s (FEV1): 67.38% ± 19.44 vs. 80.19% ± 25.24 (*p* = 0.006); the mean baseline predicted diffusion capacity of the lung for carbon monoxide (DLCO): 54.52% ± 21.29 vs. 72.42% ± 21.87 (*p* < 0.001). Patients with high COPD GOLD grade (*p* = 0.001) and high ILD GAP stage (*p* = 0.065) were more frequently assigned to proton SBRT. Notably, all patients who underwent proton SBRT were medically inoperable. 

The mean ITV (13.39 cc ± 15.07 vs. 19.44 cc ± 30.55, *p* = 0.274) and PTV (36.00 cc ± 29.07 vs. 51.28 cc ± 53.62, *p* = 0.115) were not significantly different between the groups. Dose volume histogram (DVH) analysis of the lung showed no difference between the groups with respect to the mean volume of high dose irradiated area: V_40Gy_ 4.06% ± 2.02 vs. 4.12% ± 2.38 (*p* = 0.907); and V_20Gy_ 8.90% ± 4.45 vs. 7.93% ± 3.61 (*p* = 0.233), respectively. However, significant reduction of the mean volume of low dose irradiated area was noted among the proton SBRT patients: V_10Gy_ 14.28% ± 5.52 vs. 11.33% ± 4.90 (*p* = 0.004); and V_5Gy_ 21.69% ± 7.61 vs. 14.04% ± 6.09 (*p* < 0.001), respectively.

In the matched cohort of 46 patients for photon SBRT and 28 patients for proton SBRT, no significant difference in the baseline clinical characteristics was noted: T stage (*p* = 0.667), COPD (*p* = 0.401), ILD (*p* = 0.847), mean baseline FEV1 (*p* = 0.967), or mean baseline DLCO (*p* = 0.571).

### 3.2. Survivals

The clinical outcomes of entire and matched cohort are summarized in Table 2 and are illustrated in Figure 1 and Figure 2. The 2- and 5-year rates of LC, progression-free survival (PFS), overall survival (OS), and cause-specific survival (CSS) of the entire cohort were 92.7% and 90.1%, 72.8% and 67.7%, 81.5% and 50.8%, and 90.1% and 69.2%, respectively (Figure 1).

In the entire cohort, there was no statistically significant difference in any of the clinical outcomes following either technique: LC (*p* = 0.602), PFS (*p* = 0.370), OS (*p* = 0.475), or CSS (*p* = 0.618). These findings were also similar in the matched cohort: LC (*p* = 0.472), PFS (*p* = 0.508), OS (*p* = 0.535), and CSS (*p* = 0.946) (Figure 2). 

### 3.3. Toxicity

The details of toxicity profiles are summarized in Table 3. Grade ≥ 2 toxic events were observed in 65 patients (32.2%). The most common toxic event was radiation pneumonitis (42 patients, 20.8%), followed by musculoskeletal (25 patients, 12.4%) and skin (7 patients, 3.5%) issues. There was no grade ≥ 2 cardiac or esophageal toxic event. One grade 4 toxic event occurred in an 85-year-old male patient, who developed skin necrosis needing a skin graft. This patient underwent multiport 3D-CRT SBRT with a photon beam for peripheral cT1c squamous cell carcinoma. Skin induration developed 1.5 years following SBRT, which progressed into an unhealing skin ulcer, which ultimately required surgical debridement and flap coverage 3 years following SBRT.

No statistically significant difference in grade ≥ 2 toxic events was demonstrated between the groups: radiation pneumonitis (19.6% vs. 26.4%, *p* = 0.371), musculoskeletal (13.7% vs. 5.9%, *p* = 0.264), and skin (3.6% vs. 0.0%, *p* = 0.604). The differences were also insignificant between the matched cohorts: radiation pneumonitis (30.4% vs. 21.4%, *p* = 0.398), musculoskeletal (15.2% vs. 7.1%, *p* = 0.285), and skin (6.5% vs. 0.0%, *p* = 0.468). Though insignificant, it is worthy to note that the proportion of radiation pneumonitis was reversed in the matched cohort: grade ≥ 2, from 19.6% vs. 26.4% in the entire cohort to 30.4% vs. 21.4% in the matched cohort; grade 3, from 11.9% vs. 17.6% in the entire cohort to 23.9% vs. 10.7% in the matched cohort, respectively. 

Table 4 summarizes the results of binary logistic regression analysis to identify the factors associated with grade ≥ 3 radiation pneumonitis. In the univariable analysis, poor ECOG PS (odds ratio (OR) 3.054, 95% confidence interval (95% CI) 1.231–7.580, *p* = 0.016), inoperable status (OR 8.846, 95% CI 1.165–67.144, *p* = 0.035), poor baseline FEV1 (OR 3.818, 95% CI 1.062–13.731, *p* = 0.040), and poor baseline DLCO (OR 4.980, 95% CI 1.636–15.162, *p* = 0.005) were significantly associated with radiation pneumonitis. In the multivariable analysis, poor ECOG PS (OR 3.162, 95% CI 1.215–8.226, *p* = 0.018) and poor baseline DLCO (OR 3.995, 95% CI 1.259–12.675, *p* = 0.019) were significantly associated with radiation pneumonitis.

## 4. Discussion

The current study is a retrospective study from a single tertiary institution that evaluated the clinical outcomes following either or photon or proton SBRT for treating early-stage lung cancer patients. Both photon and proton SBRT resulted in favorable clinical outcomes with acceptable toxicity profiles when compared with the previous reports (Table 5). Proton SBRT, when compared with photon SBRT, was capable of generating more favorable dose profiles, typically reducing the low dose irradiated lung volume. Though the baseline patient characteristics were different in that high-risk patients were assigned more frequently to proton SBRT in the current study, there was no significant difference in the clinical outcomes or toxicity profiles between groups.

The clinical efficacy of SBRT for early-stage lung cancer has been proven through several prospective studies [18,19,20,21,22,23,24]. Following SBRT, the LC rates at 2 or 5 years ranged from 85% to 96%, and grade ≥ 3 toxic events were observed in 2.6%–28% of the patients in these prospective studies (Table 5). Through the current study, following either photon-or proton SBRT, the 5-year rates of LC and OS were 85.1% and 91.0%, and 54.1% and 51.6%, respectively, and grade ≥ 3 toxic events occurred in 15.8% and 18.0%, respectively. These clinical outcomes were comparable to those of the above-mentioned prospective studies.

While photon SBRT for early-stage NSCLC has been thoroughly studied, the benefit of proton SBRT over photon SBRT has not been clearly identified. Bayasgalan et al. [25], who compared the dosimetric profiles of the proton beam and modern photon RT for stage I lung cancer patients, reported dosimetric advantages of proton beam RT with respect to the mean lung dose, lung V5 and V10, mean heart dose, and heart V5 and V10. Kadoya et al. [9] did a similar comparative study and concluded that proton beam therapy significantly reduced the low-dose irradiation volume, and was more advantageous in the management of large tumors. The International Particle Therapy Cooperative Group published the consensus statements of proton beam therapy in the management of NSCLC [26], which summarized that proton SBRT delivers excellent dose distributions with high local control and survival. Though photon SBRT via modernized techniques also can generate similarly favorable dose profiles, proton SBRT, could be more favorable in treating the lesions, such as the centrally located lesions or lesions close to the brachial plexus, in which the dosimetric constraints could not be easily fulfilled by photon SBRT.

In the current study, the patients who were known to have a high risk of radiation pneumonitis, or as having poor baseline PFT results or multiple comorbidities (especially ILD) [27,28], underwent proton SBRT more frequently than photon SBRT. The incidences of radiation pneumonitis, however, were not different following either modality. Moreover, in the binary logistic regression analysis on grade ≥3 radiation pneumonitis, worse ECOG PS and poor baseline DLCO, but not SBRT modality, were significantly associated with radiation pneumonitis in the multivariable analysis. Though statistically insignificant, proton SBRT resulted in less frequent musculoskeletal and skin toxicity when compared to photon SBRT. Chest wall toxicity after SBRT is correlated with dose delivered to the chest wall [29], and proton SBRT is known to reduce the dose to the chest wall when compared to photon SBRT [30]. Based on the authors’ observations, it is difficult to propose the optimal indications of proton SBRT. However, helpful hints could be given for defining the candidates for proton SBRT.

The current study suffered from the limitation of selection bias, which stemmed from being retrospective in nature. Additionally, due to the small number of patients included in the proton SBRT arm, the outcomes could have been overestimated, or significance could have been masked. Additionally, it is noteworthy that though patients were thoroughly reviewed with CT and PET-CT before SBRT, a significant proportion of patients had unconfirmed pathology (32.2%). This implies that some fraction of our patients in the current study could have had either benign lesions or small cell carcinoma histology. Therefore, caution should be used when interpreting our observations. However, it is noteworthy in that the current study had a few merits: first, we accrued a relatively large homogeneous patient cohort from a single tertiary institution; second, consistent SBRT policies of contouring, planning, and dose schedule were applied to all patients; and third, efforts to increase the comparability by propensity score matching was utilized. Considering all these limitations and merits, both proton and photon SBRT resulted in favorable clinical outcomes with acceptable toxicity profiles when compared with previous reports (Table 5).

## 5. Conclusions

Photon and proton SBRT resulted in comparable oncologic outcomes with similar toxicity profiles, though the patients at high risk of developing lung toxicity underwent proton SBRT more frequently. Based on current observations, it could be speculated that proton SBRT should be considered for the patients at high-risk of radiation pneumonitis. A further prospective randomized study with a large sample, however, would be needed to prove our findings and to identify the ideal candidates for each modality.

## Figures and Tables

**Figure 1 cancers-14-04152-f001:**
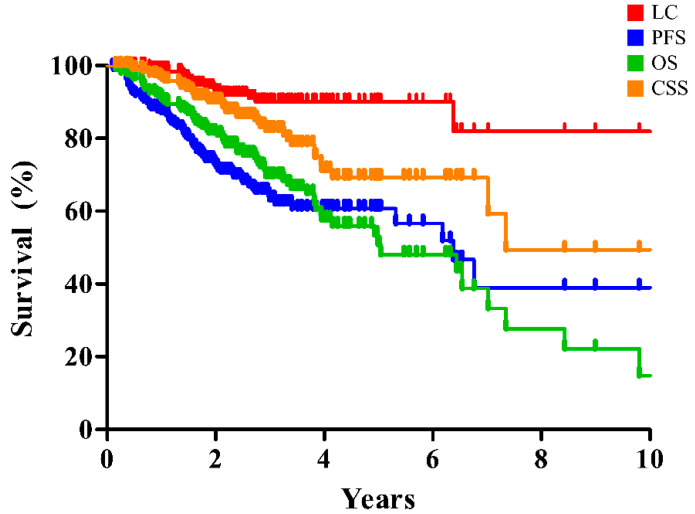
Clinical outcomes of SBRT for early-stage lung cancer in the entire cohort. SBRT, stereotactic body radiation therapy; LC, local control; PFS, progression-free survival; OS, overall survival; CSS, cause-specific survival.

**Figure 2 cancers-14-04152-f002:**
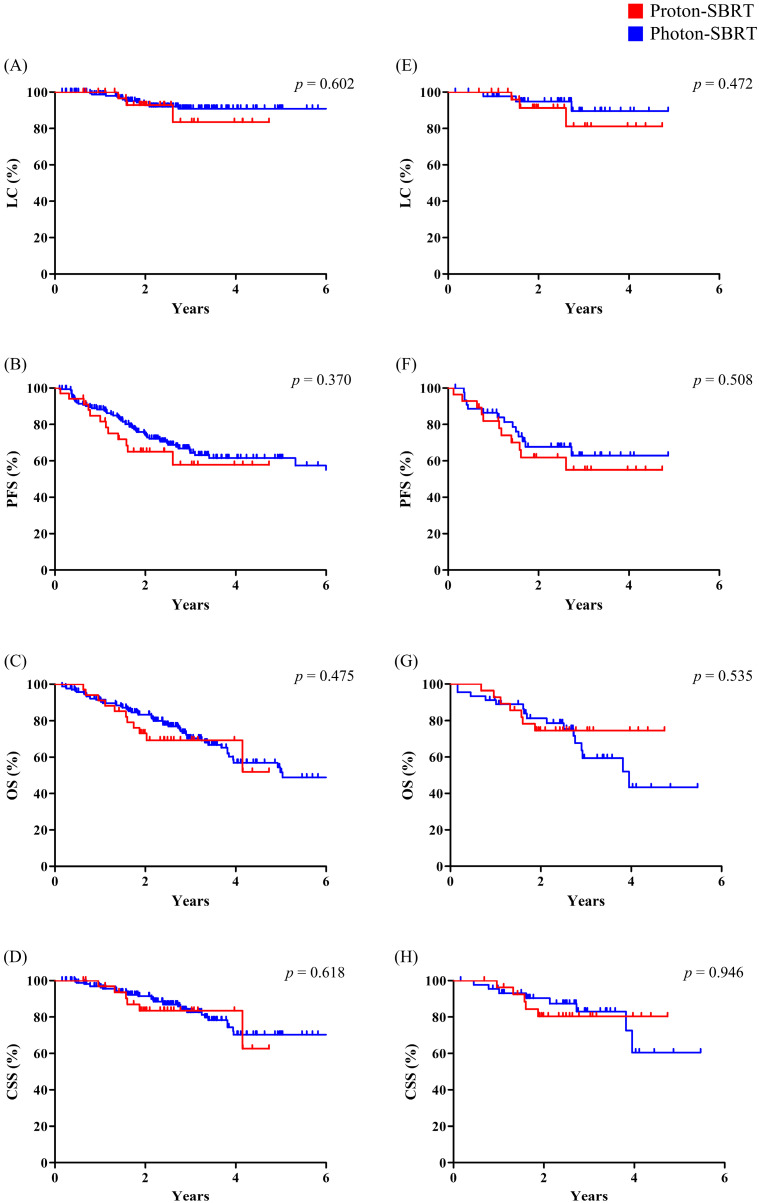
Comparison of clinical outcomes between proton SBRT and photon SBRT. (**A**–**D**) Entire cohort; (**E**–**H**) matched cohort. SBRT, stereotactic body radiation therapy; LC, local control; PFS, progression-free survival; OS, overall survival; CSS, cause-specific survival.

**Table 1 cancers-14-04152-t001:** Baseline clinical and treatment characteristics.

	Entire Cohort (*N* = 202)				Matched Cohort (*N* = 74)		
	Overall (*n* = 202)	Photon SBRT (*n* = 168)	Proton SBRT (*n* = 34)	*p* value	Photon SBRT (*n* = 46)	Proton SBRT (*n* = 28)	*p* value
Age (years, median, IQR)	75 (70–79)	76 (70–79)	72.5 (68–76)	0.288	75 (70–78)	73 (68–76)	0.411
Sex				0.963			0.667
Male	161 (79.7%)	134 (79.8%)	27 (79.4%)		38 (82.6%)	22 (78.6%)	
Female	41 (20.3%)	34 (20.2%)	7 (20.6%)		8 (17.4%)	6 (21.4%)	
ECOG PS				0.151			0.113
0–1	167 (82.7%)	136 (81.0%)	31 (91.2%)		35 (76.1%)	26 (92.9%)	
2-	35 (17.3%)	32 (19.0%)	3 (8.8%)		11 (23.9%)	2 (7.1%)	
Pathology				0.241			0.589
Adenocarcinoma	77 (38.1%)	66 (39.3%)	11 (32.4%)		16 (34.8%)	10 (35.7%)	
Squamous cell carcinoma	47 (23.3%)	38 (22.6%)	9 (26.5%)		12 (26.1%)	8 (28.6%)	
Other	13 (6.4%)	13 (7.7%)	0 (0.0%)		3 (6.5%)	0 (0.0%)	
Unproven	65 (32.2%)	51 (30.4%)	14 (41.2%)		15 (32.6%)	10 (35.7%)	
Location				0.176			0.479
LLL	34 (16.8%)	25 (14.9%)	9 (26.5%)		5 (10.9%)	7 (25.0%)	
LUL	55 (27.2%)	48 (28.6%)	7 (20.6%)		12 (26.1%)	6 (21.4%)	
RLL	41 (20.3%)	31 (18.5%)	10 (29.4%)		11 (23.9%)	8 (28.6%)	
RML	7 (3.5%)	6 (3.6%)	1 (2.9%)		3 (6.5%)	1 (3.6%)	
RUL	65 (32.2%)	58 (34.5%)	7 (20.6%)		15 (32.6%)	6 (21.4%)	
Tumor size (mm, mean ± SD)	21.77 ± 8.52	21.62 ± 8.22	22.50 ± 10.00	0.584	22.00 ± 9.60	23.32 ± 9.96	0.573
T stage				0.178			0.667
T1	170 (84.2%)	144 (85.7%)	26 (76.5%)		38 (82.6%)	22 (78.6%)	
T2	32 (15.8%)	24 (14.3%)	8 (23.5%)		8 (17.4%)	6 (21.4%)	
COPD	85 (42.1%)	61 (36.3%)	24 (70.6%)	<0.001	25 (54.3%)	18 (64.3%)	0.401
COPD GOLD grade				0.001			0.645
Grade 1	22 (10.8%)	18 (10.7%)	4 (11.8%)		6 (13.0%)	3 (10.7%)	
Grade 2	46 (22.7%)	33 (19.6%)	13 (38.2%)		13 (28.3%)	12 (42.9%)	
Grade 3	16 (7.9%)	9 (5.4%)	7 (20.6%)		6 (13.0%)	3 (10.7%)	
Grade 4	1 (0.5%)	1 (0.6%)	0 (0.0%)		0 (0.0%)	0 (0.0%)	
ILD	25 (12.4%)	18 (10.7%)	7 (20.6%)	0.149	9 (19.6%)	6 (21.4%)	0.847
ILD GAP stage				0.065			0.845
Stage 1	6 (3.0%)	6 (3.6%)	0 (0.0%)		1 (2.2%)	0 (0.0%)	
Stage 2	17 (8.4%)	11 (6.5%)	6 (17.7%)		7 (15.2%)	5 (17.9%)	
Stage 3	2 (1.0%)	1 (0.6%)	1 (2.9%)		1 (2.2%)	1 (3.6%)	
Baseline FEV1 (% predicted, mean ± SD)	77.85 ± 24.74	80.19 ± 25.24	67.38 ± 19.44	0.006	71.46 ± 22.71	71.25 ± 16.45	0.967
Baseline DLCO (% predicted, mean ± SD)	68.98 ± 22.82	72.42 ± 21.87	54.52 ± 21.29	<0.001	60.28 ± 17.51	57.68 ± 21.40	0.571
Operability				<0.001			0.001
Operable	47 (23.3%)	47 (28.0%)	0 (0.0%)		13 (28.3%)	0 (0.0%)	
Inoperable	155 (76.7%)	121 (72.0%)	34 (100.0%)		33 (71.7%)	28 (100.0%)	
SBRT technique				<0.001			<0.001
3D-CRT	130 (64.4%)	130 (77.4%)	0 (0.0%)		29 (63.0%)	0 (0.0%)	
IMRT	38 (18.8%)	38 (22.6%)	0 (0.0%)		17 (37.0%)	0 (0.0%)	
Passive scattering	4 (2.0%)	0 (0.0%)	4 (11.8%)		0 (0.0%)	4 (14.3%)	
IMPT	30 (14.8%)	0 (0.0%)	30 (88.2%)		0 (0.0%)	24 (85.7%)	
Respiratory motion control				<0.001			<0.001
Free breathing	187 (92.6%)	167 (99.4%)	20 (58.8%)		46 (100.0%)	16 (57.1%)	
Gating	1 (0.5%)	0 (0.0%)	1 (2.9%)		0 (0.0%)	1 (3.6%)	
DIBH	14 (6.9%)	1 (0.6%)	13 (38.2%)		0 (0.0%)	11 (39.3%)	
Dosimetric parameters							
ITV (cc, mean ± SD)	14.44 ± 18.73	13.39 ± 15.07	19.44 ± 30.55	0.274	12.48 ± 13.63	19.68 ± 32.38	0.189
PTV (cc, mean ± SD)	38.57 ± 34.74	36.00 ± 29.07	51.28 ± 53.62	0.115	34.38 ± 27.77	52.61 ± 55.59	0.064
Lung V_40Gy_ (%, mean ± SD)	4.07 ± 2.66	4.06 ± 2.72	4.12 ± 2.38	0.907	4.16 ± 3.68	4.49 ± 2.40	0.674
Lung V_20Gy_ (%, mean ± SD)	8.74 ± 4.33	8.90 ± 4.45	7.93 ± 3.61	0.233	9.28 ± 5.51	8.51 ± 3.45	0.513
Lung V_10Gy_ (%, mean ± SD)	13.78 ± 5.52	14.28 ± 5.52	11.33 ± 4.90	0.004	14.72 ± 6.09	12.07 ± 4.68	0.053
Lung V_5Gy_ (%, mean ± SD)	20.40 ± 7.90	21.69 ± 7.61	14.04 ± 6.09	<0.001	22.02 ± 7.79	14.90 ± 5.86	<0.001

SBRT, stereotactic body radiation therapy; ECOG PS, Eastern Cooperative Oncology Group performance status; LLL, left lower lobe; LUL, left upper lobe; RLL, right lower lobe; RML, right middle lobe; RUL, right upper lobe; COPD, chronic obstructive pulmonary disease; GOLD, Global Initiative for Chronic Obstructive Lung disease; ILD, interstitial lung disease; GAP, gender (G), age (A), and 2 lung physiology variables (P); FEV1, forced expiratory volume in 1 s; DLCO, diffusion capacity of lung for carbon monoxide; 3D-CRT, three dimensional conformal radiation therapy; IMRT, intensity modulated radiation therapy; IMPT, intensity modulated proton therapy; DIBH, deep inspiration breath hold; ITV, internal target volume; PTV, planning target volume; IQR, interquartile range; SD, standard deviation.

**Table 2 cancers-14-04152-t002:** Clinical outcomes of SBRT.

		Entire Cohort (*N* = 202)				Matched Cohort (*N* = 74)		
		Overall (*n* = 202)	Photon SBRT (*n* = 168)	Proton SBRT (*n* = 34)	*p* value	Photon SBRT (*n* = 46)	Proton SBRT (*n* = 28)	*p* value
LC	2-year	92.7%	92.8%	92.8%	0.602	94.9%	91.3%	0.472
	5-year	90.1%	90.8%	83.6%		89.6%	81.1%	
PFS	2-year	72.8%	74.4%	65.0%	0.370	67.7%	61.9%	0.508
	5-year	60.7%	61.6%	57.8%		62.9%	55.0%	
OS	2-year	81.5%	83.3%	73.1%	0.475	81.3%	74.5%	0.535
	5-year	50.8%	51.7%	51.9%		43.4%	74.5%	
CSS	2-year	90.1%	91.5%	83.5%	0.618	90.4%	80.4%	0.946
	5-year	69.2%	70.3%	62.6%		60.5%	80.4%	

SBRT, stereotactic body radiation therapy; LC, local control; PFS, progression-free survival; OS, overall survival; CSS, cause-specific survival.

**Table 3 cancers-14-04152-t003:** Toxic events after SBRT.

	Entire Cohort (*N* = 202)							Matched Cohort (*N* = 74)						
	Photon SBRT (*n* = 168)			Proton SBRT (*n* = 34)			*p* value	Photon SBRT (*n* = 46)			Proton SBRT (*n* = 28)			*p* value
	G2	G3	G4	G2	G3	G4		G2	G3	G4	G2	G3	G4	
Radiation pneumonitis	13 (7.7%)	20 (11.9%)	0 (0.0%)	3 (8.8%)	6 (17.6%)	0 (0.0%)	0.371 *	3 (6.5%)	11 (23.9%)	0 (0.0%)	3 (10.7%)	3 (10.7%)	0 (0.0%)	0.398 *
Musculoskeletal	17 (10.1%)	6 (3.6%)	0 (0.0%)	2 (5.9%)	0 (0.0%)	0 (0.0%)	0.264 **	6 (13.0%)	1 (2.2%)	0 (0.0%)	2 (7.1%)	0 (0.0%)	0 (0.0%)	0.285 **
Skin	3 (1.8%)	3 (1.8%)	1 (0.6%)	0 (0.0%)	0 (0.0%)	0 (0.0%)	0.604 **	2 (4.3%)	1 (2.2%)	0 (0.0%)	0 (0.0%)	0 (0.0%)	0 (0.0%)	0.468 **

SBRT, stereotactic body radiation therapy; G, grade. * Chi-square test, ** Fisher’s exact test.

**Table 4 cancers-14-04152-t004:** Univariable and multivariable binary logistic regression analysis for grade 3 or higher radiation pneumonitis.

	Univariable		Multivariable	
	OR (95% CI)	*p*-value	OR (95% CI)	*p*-value
Sex (Male)	7.606 (1.002–57.709)	0.054		
Age (>70)	1.400 (0.498–3.934)	0.523		
ECOG PS (2 or higher)	3.054 (1.231–7.580)	0.016	3.162 (1.215–8.226)	0.018
Smoking History (Yes)	2.349 (0.669–8.252)	0.183		
COPD (Yes)	1.444 (0.633–3.297)	0.383		
ILD (Yes)	2.479 (0.886–6.937)	0.084		
T stage (T2)	1.731 (0.635–4.718)	0.284		
Operability (Inoperable)	8.846 (1.165–67.144)	0.035	7.204 (0.929–55.863)	0.059
Baseline FEV1 (<40%)	3.818 (1.062–13.731)	0.040		
Baseline DLCO (<40%)	4.980 (1.636–15.162)	0.005	3.995 (1.259–12.675)	0.019
Respiratory motion control (No)	0.611 (0.162–2.310)	0.468		
Treatment modality (Photon)	0.631 (0.233–1.710)	0.365		

OR, odds ratio; CI, confidence interval; ECOG PS, Eastern Cooperative Oncology Group performance status; COPD, chronic obstructive pulmonary disease; ILD, interstitial lung disease; FEV1, forced expiratory volume in 1 s; DLCO, diffusion capacity of lung for carbon monoxide.

**Table 5 cancers-14-04152-t005:** Summary of prospective studies of SBRT for early-stage lung cancer, along with the current study.

Article	Patients	Stage	Dose	Follow-up	LC	OS	Toxicity
Hoyer et al. [18]	40	Stage I	45 Gy in 3 fractions	29 months	2-year, 85%	2-year, 47%	Grade ≥ 2, 48%
Fakiris et al. [19]	70	T1-T2	60–66 Gy in 3 fractions	50 months	3-year, 88.1%	3-year, 42.7%	Grade ≥ 3, 15.7%
Baumann et al. [20]	57	T1-T2	45 Gy in 3 fractions	35 months	3-year, 92%	3-year, 60%	Grade ≥ 3, 28%
Timmerman et al. [21]	55	T1-T2	54 Gy in 3 fractions	34 months	3-year, 91%	3-year, 56%	Grade ≥ 3, 16%
Timmerman et al. [22]	33	T1-T2	54 Gy in 3 fractions	48.1 months	4-year, 96%	4-year, 57%	Grade ≥ 3, 8%
Nagata et al. [23]	100 (inoperable)	T1	48 Gy in 4 fractions	47 months	3-year, 87.3%	3-year, 59.9%	Grade ≥ 3, 12.5%
	64 (operable)	T1	48 Gy in 4 fractions	67 months	3-year, 85.4%	3-year, 76.5%	Grade ≥ 3, 6.2%
Videtic et al. [24]	39	T1-T2	34 Gy in 1 fraction	3.5 years	5-year, 89.4%	5-year, 29.6%	Grade ≥ 3, 2.6%
	45	T1-T2	48 Gy in 4 fractions	4.0 years	5-year, 93.2%	5-year, 41.1%	Grade ≥ 3, 11.1%
Current study	168 (Photon)	T1-T2	60 Gy in 4 fractions	2.7 years	5-year, 90.8%	5-year. 51.7%	Grade ≥ 3, 16.7%
	34 (Proton)	T1-T2	60 Gy in 4 fractions	2.2 years	5-year, 83.6%	5-year, 51.9%	Grade ≥ 3, 17.6%

SBRT, stereotactic body radiation therapy; LC, local control; OS, overall survival.

## Data Availability

The data are not publicly available due to ethical issues.

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
