# Peer review of "Clinical Outcomes Following Proton and Photon Stereotactic Body Radiation Therapy for Early-Stage Lung Cancer"

_cancers, 2022, doi:10.3390/cancers14174152_

Round 1
Reviewer 1 Report
This is a retrospective study of non-small cell lung patients treated with either photon or proton based radiotherapy.
The information is important.
The results show no detectable difference in clinical outcomes for patients treated with either modality. The study was not powered to detected superiority or non-inferiority.
Thus, the authors' conclusions are not justified by the results. The study shows that proton based radiotherapy can be considered to be a safe treatment option for inoperable patients.
There is no data to indicate or suggest superiority over photon based radiotherapy.
Line 30: Term "preferably" should be deleted
Line 254: Delete "until yet"
Line 296: Delete "preferably"
Reviewer 2 Report
The comparison between proton or photon radiotherapy for lung cancer is very important issue. However, there are some points to revise before publishing.
1. I understand that there are many patients with difficult of cyto-pathologic conformation. Therefore, non-small cell lung cancer as you said in abstract is not correct.
2. How did you decide medically inoperable?
3. There are many degrees of COPD and ILD, and the decision of treatment will change because of those degree. Could you show us these detail data?
4. There were 38% patients with DIBH in proton group. How did you decide respiratory motion control?
5. I would like to know changes respiratory function after treatment. Could you show?
6. Please show us details of toxicities about pulmonary?
Round 2
Reviewer 1 Report
Please remove word preferably from abstract. It is not supported by the data
Author Response
Please remove word preferably from abstract. It is not supported by the data
Thank you for your comment. We have changed the sentence in Simple summary as your suggestion.
(Line 13-16) Proton-SBRT could be considered for patients with high risk of radiation pneumonitis.
Reviewer 2 Report
Authors answered my comment properly. So, I do not have adding comment for this article. 
Author Response
Authors answered my comment properly. So, I do not have adding comment for this article.
Thank you for reviewing our manuscript. The comments of the reviewer was very helpful in improving the manuscript. We hope that the revised manuscript is suitable for publication.